# SARS-CoV-2 Transmission in the Military during the Early Phase of the Pandemic—A Systematic Analysis

**DOI:** 10.3390/ijerph19127418

**Published:** 2022-06-16

**Authors:** Sylvia Xiao Wei Gwee, Pearleen Ee Yong Chua, Junxiong Pang

**Affiliations:** 1Saw Swee Hock School of Public Health, National University of Singapore, National University Health System, Singapore 117549, Singapore; ephsgxw@nus.edu.sg (S.X.W.G.); e0320414@u.nus.edu (P.E.Y.C.); 2Centre for Infectious Disease Epidemiology and Research, National University of Singapore, Singapore 117549, Singapore

**Keywords:** COVID-19, military, transmission, exposure, clinical characteristics

## Abstract

Militaries worldwide have been affected by COVID-19 pandemic. However, the impact and epidemiological characteristics of transmission during the early phase of the pandemic is not well-studied. This study aims to systematically estimate the baseline incidence of COVID-19 in the military worldwide and identify the potential risk factors of transmission and clinical characteristics of the cases. English and Chinese literature reporting COVID-19 cases in military worldwide published on four electronic databases (*PubMed, Scopus, EMBASE*, and *CKNI*) through 28 May 2021 were systematically screened and synthesized qualitatively. Forty-six studies involving at least 711,408 military personnel in 17 countries were synthesized. Low incidence of cases was observed in the military with pooled COVID-19 incidence of 0.19% (95%CI: 0.00–9.18%). We observed a higher incidence among those (1) with overseas exposure (39.85%; 95%CI: 0.00–95.87%) rather than local exposure (3.03%; 95%CI: 0.00–12.53%), (2) who were on either local/overseas military deployment (26.78%; 95%CI: 0.00–71.51%) as compared to those not deployed (4.37%; 95%CI: 0.00–17.93%), and (3) on overseas military deployment (39.84%; 95%CI: 0.00–95.87%) as compared to local military deployment (3.03%; 95%CI: 2.37–3.74%). The majority of the cases were symptomatic (77.90% (95%CI: 43.91–100.00%)); hospitalization and mortality rates were low at 4.43% (95%CI: 0.00–25.34%) and 0.25% (95%CI: 0.00–0.85%), respectively; and headache, anosmia, ageusia, myalgia, nasal congestion, and cough were the most commonly observed symptoms. Overseas and local deployment were observed to have higher risk of SARS-CoV-2 transmission. Sustainable, active SARS-CoV-2 surveillance strategies are crucial to detect and contain transmission early during military deployments.

## 1. Introduction

Reports on an outbreak of pneumonia caused by an unknown etiological agent first broke out in Wuhan, China, during late December 2019. While the World Health Organization declared the outbreak a Public Health Emergency of International Concern on 30 January 2020, the coronavirus disease 2019 (COVID-19), due to severe acute respiratory syndrome coronavirus 2 (SARS-CoV-2), was only declared a pandemic on 11 March 2020 [1].

The military is characterized by attributes that are advantageous for managing such disease outbreaks: crisis-management capacities, ability to execute missions in sub-optimal environments, logistic resources for deployment, and the ability to mobilize large forces in risky situations within or outside the country [2]. While some militaries have service members with substantial emergency-response background and public health expertise, those without can also utilize its national command network, pool of disciplined manpower, and logistical support to supplement civilian frontline services [3].

Gibson-Fall outlined three trends of military involvement of various degrees that emerged from the early phase of the COVID-19 pandemic [4]. Militaries, such as in Canada [5,6], provided minimal technical support in their niches, such as transportation, supply chain, and border control, to support civilian response. Countries such as the United States (U.S.) [7], Singapore [8], and China [9] utilized a blended military–civil response, which saw military support extending to organizations, logistics, border control, testing, quarantine, source investigations, lockdown enforcement, and emergency field hospitals. Alternatively, the military can lead the pandemic response in all aspects of planning, coordination, and execution, as with Pakistan [10].

As the military’s operational support are at the frontlines, they are inevitably subjected to a heightened risk of pathogen exposure. While the occurrence of disease outbreaks in military due to overseas deployment and disaster management is well-established [11], the impact and epidemiological characteristics of transmission during the early phase of the SARS-CoV-2 pandemic is not well-studied. Assessing the military’s risk during this pandemic could be a useful reference for novel respiratory virus outbreaks in the future and play a role in guiding their preparation. This study aims to estimate the incidence of SARS-CoV-2 infections in militaries during the early phase of the pandemic and identify potential risk factors behind transmission and their key clinical characteristics. 

## 2. Materials and Methods

### 2.1. Search Strategy and Screening

This systematic analysis was performed in accordance with the Preferred Reporting Items for Systematic Reviews and Meta-Analyses (PRISMA) guidelines [12]. English and Chinese literature that reported COVID-19 cases in military worldwide were extracted from four electronic databases, namely *PubMed, Scopus, EMBASE*, and *CKNI,* on 13 April 2021. Key search terms representing variations of COVID-19 and military personnel were used—“SARS-CoV-2”, “2019-nCoV”, “COVID-19”, “2019 novel coronavirus”, “army”, “soldiers”, “troop”, and “military” were used in the systematic search. For Chinese database *CKNI*, search terms “新冠病毒”, “新型冠状病毒”, “新冠肺炎”, “军队”, “士兵”, “武装部队”, “部队”, and “军人” were used. To ensure the relevance of this study, the search was updated to include newer publications using the same strategy on 28 May 2021.

Inclusion of identified publications followed the following criteria:(1)Population: Official government-linked armed forces consisting of active serving personnel in the Army, Navy, Air Force, and other relevant units;(2)Outcome: Confirmed COVID-19 cases or SARS-CoV-2 infections in the military;(3)Intervention: All settings that involve military personnel—local and overseas deployment on military vehicles, hospitals, communities, or within military schools or training centers;(4)Study Design: All reliable sources regardless of article type;(5)Comparator: Not applicable.

### 2.2. Data Extraction

Seventy-two data fields constituting of six main domains, namely study details, overall military population demographics, incidence of cases and its testing platforms, case demographics, exposures, and clinical characteristics, were extracted from each study to Microsoft Excel 2016. Outcome measures include number of cases, total personnel involved, testing coverage, mode of diagnosis, incidence by RT-PCR/serology, and test kits used. Exposures were grouped into the following categories: local or overseas, deployment, duration of deployment, recreational activity, close contact (secondary transmission), healthcare, and others. Clinical characteristics comprised the number of symptomatic/asymptomatic cases, total cases with symptom-related data available, breakdown of individual symptoms, number of hospitalizations, and deaths. A confirmed case was defined as SARS-CoV-2 infection regardless of the development of COVID-19 disease.

### 2.3. Data Analysis

Apart from qualitative synthesis of the data, meta-analysis was explored to gain additional insights. Incidence of the exposure, outcomes and clinical characteristics, and their corresponding 95% confidence intervals were pooled using the inverse variance heterogeneity (IVhet) model [13]. The model is a modification of the fixed-effects model that accounts for between-study heterogeneity while retaining the individual weight of studies [14]. Freeman–Tukey double arcsine transformation was used to avoid giving weight to studies with estimates that are too skewed. Forest plots were generated for graphical representations. I^2^ statistic values were calculated to quantify degree of heterogeneity among studies that was not attributable to chance; values of 25–50% suggest heterogeneity, and values of >50% indicate substantial heterogeneity. All meta-analyses were conducted with MetaXL meta-analyses software (version 5.3, EpiGear International, Sunrise Beach, Australia).

Studies including case reports and case series without military population for denominator were excluded from the meta-analyses; those constituting less than five patients were also excluded from all meta-analyses of clinical characteristics [15]. Subgroup analysis of incidence was conducted for (1) deployment or no deployment, (2) reporting of case from local or overseas setting, and (3) deployment to a local or overseas setting.

## 3. Results

### 3.1. Published Literature

The initial database searches identified 3307 studies. Following the removal of 547 duplicates, 2760 studies were screened for their titles and abstracts. From 141 studies shortlisted in the primary screen (Figure 1), 46 studies in English language were selected and synthesized in this systematic analysis (Table 1, Table 2, Table 3, Appendix A). Only 36 of the included studies were used in various meta-analyses (Appendix A), as the remaining ten had a potential overlap in cases due to the period of reporting. [16,17,18,19,20,21,22,23,24,25].

#### 3.1.1. Study Characteristics

Included studies (*n* = 46) were mostly from the U.S. (*n* = 22). The remaining studies originated from Israel (*n* = 5), Switzerland (*n* = 3), France (*n* = 2), the United Kingdom (*n* = 2), Belgium, Bolivia, Brazil, Canada, Djibouti, India, Italy, Norway, Philippines, South Korea, Sri Lanka, and Tunisia (1 each) (Table 1). Included articles comprised research articles (*n* = 27), research letters (*n* = 5), news articles (*n* = 5), letters to the editor (*n* = 2), CDC morbidity and mortality weekly report (*n* = 3), commentary (*n* = 1), fast facts (*n* = 1), rapid communication (*n* = 1), and short report (*n* = 1). Using the maximum cases mentioned but excluded in each study’s original analysis, there was likely a maximum of 79,725 cases reported from 17 countries between January 2020 and 7 June 2021. Most of the cases were attributed to a research letter on the U.S. military dated through 2 November 2020 [25].

After accounting for potential overlap in cases, this study utilized 36 articles from 17 countries—the U.S. (*n* = 17), Israel (*n* = 2), Switzerland (*n* = 2), and the United Kingdom (*n* = 2) and one each from Belgium, Bolivia, Brazil, Canada, Djibouti, France, India, Italy, Norway, Philippines, South Korea, Sri Lanka, and Tunisia [26,27,28,29,30,31,32,33,34,35,36,37,38,39,40,41,42,43,44,45,46,47,48,49,50,51,52,53,54,55,56,57,58,59,60,61]. The Israel-based study by Talmy et al. was solely used for the meta-analysis of specific symptoms since it was the only Israeli study reporting symptoms breakdown [58]. There were at least 24,930 males reported by 19 studies, comprising 50.20% to 93.10% of these study populations (Table 1). The mean/median age reported by nine studies ranged from 19.1 to 45.1 years old; eight were below 34 years old, while the sole study with a higher mean of 45.1 years old was based in a hospital setting. Five case reports had patients ranging from 21 to 36 years old. Settings where positive cases were detected were available for 28 studies—military hospitals/treatment facilities (*n* = 8), aircraft carriers (*n* = 2), recruit schools (*n* = 2), army bases (*n* = 4), quarantine facilities (*n* = 5), deployment at field hospital/hospital ship (*n* = 2), and arrival testing—on return from deployment (*n* = 2)/at deployment site (*n* = 1), deployment at long-term care facilities (*n* = 1), and air evacuation to medical facilities (*n* = 1).

**Table 1 ijerph-19-07418-t001:** Study details and population demographics of included studies.

Study ^α^	Publication	Study Type	Country	Study Period ^&^	Population ^@^	Cases Only	Male (%) ^^^	Age ^%,^^	Setting
Pirnay, J.P. (2020) [51]	Research Article	Retrospective cohort	Belgium	May 2020	Belgian soldiers of “Mobile Education and Training Team” that trained special intervention company of Nigerian soldiers, undergoing arrival testing	N	70 (100%)		Arrival testing upon return from deployment
Escalera-Antezana, J.P. (2020) [33]	Research Letter	Letter	Bolivia	Unknown; 2 months	Military personnel of Bolivia; surveillance testing	N			
Pasqualotto, A.C. (2021) [50]	Research Article	Cross-sectional	Brazil	23–25 July 2020	Military police in ten cities of Rio Grande do Sul—Porto Alegre, Caxias do Sul, Canoas, Pelotas, Santa Maria, Passo Fundo, Uruguaiana, Santa Cruz do Sul, Ijui, and Lajeado—who had no previous confirmed COVID-19	N	1292 (81.2%)	34 (8) ^e^	
Halladay, J. (2020) [35]	Fast Facts	Information sheet	Canada	April–7 July 2020	Canadian Armed Forces deployed to long-term care facilities	N			Deployment to long-term care facilities
Elhakim, M. (2020) * [31]	Research Article	Pandemic response	Djibouti		Foreign military contingent deployed to Djibouti	Y	1 (100%)		Arrival testing
Paleiron, N. (2021) [49]	Research Article	Cross-sectional	France	April 2020	Charles de Gaulle sailors under outbreak investigation	N	1466 (87%)	28 (23–35)	Aircraft carrier
*Chassery, L. (2021)* [17]	*Research Article*	*Retrospective outbreak analysis*	*France*	*April 2020*	*French Navy sailors on Charles de Gaulle*	*Y*			*Aircraft carrier*
Joshi, R.K. (2020) [37]	Research Letter	Prospective cohort	India	30 May–12 July 2020	Indian security forces personnel placed in 14-day quarantine after return from leave	N			Quarantine facility
Sasongko, S. (2021) ^#^ [52]	Research Article	Cross-sectional	Indonesia	15 August–15 November 2020	Inpatients of military/police occupation at Dustira Army Hospital with suspected COVID-19. Only those who had an RT-PCR swab and complete patient data included	N			Dustira Army Hospital
Nitecki, M. (2021) [47]	Research Article	Retrospective cohort	Israel	26 March–2 August 2020	Israel soldiers deemed eligible for COVID-19 testing by the ICC, including those voluntarily calling to report symptoms or a suspected exposure or those actively addressed following an epidemiological investigation.	N	14398 (59.1%)	20.5 (19.6–22.4)	
*Segal, D. (2020)* [22]	*Research Article*	*Pandemic response*	*Israel*	*26 February–19 April 2020*	*Israel Defense Force Northern Command*	*N*	*81.34%*	*21.29 (4.06) ^c^*	
Talmy, T. (2021) [58]	Letter to the Editor	Case series	Israel	20 March–5 May 2020	Israel Defense Force (IDF) soldiers with positive SARS-CoV-2 test admitted to rehabilitation center by IDF Medical Corps	Y	84 (70.6%)	21 (19–25)	Rehabilitation center
*Segal, D. (2021)* [21]	*Research Article*	*Pandemic response; vaccination*	*Israel*	*1 March 2020* *–18 February 2021*	*Individuals who served in 70 military units that have been allocated to three vaccination stations (for vaccination)*	*N*	*13290 (71.1%)*	*22.77 (1.35) ^f^*	
*Tsur, A. (2021)* [24]	*Research Article*	*Screening policy/Pandemic response*	*Israel*	*1 April* *–14 May 2020*	*Israel Defense Forces soldiers from 13 units*	*N*	*769 (78%)*	*18–52 ^d^*	
Di Nunno, D. (2020) [30]	Research Article	Prospective case series	Italy	16 March–4 May 2020	Italian servicemen infected by SARS-CoV-2 during overseas deployment/military operations hospitalized in non-intensive wards	Y	75 (92.6%)	45.1 (10.4)	Celio Military Hospital
Borud, E.K. (2021) [28]	Research Article	Prospective cohort	Norway	6 weeks from 19–27 April 2020	Norwegian conscripts who just enrolled into military training and followed for 6 weeks	N			Army Base
Velasco, J.M. (2020) [60]	Research Article	Prospective cohort	Philippines	14 April–15 August 2020	Philippines military personnel under investigation for COVID-19, patients seeking clinical care and showing signs of COVID-19-like illness, or asymptomatic patients as part of contact tracing procedures.	N			V Luna Medical Center (tertiary care, teaching hospital under the Armed Forces of the Philippines Health Service Command), other military hospitals in Metro Manila
Oh, H.S. (2020) ^$^ [48]	Commentary	Pandemic response	South Korea	January–End June 2020	Confirmed cases in military population of South Korea	N	54 (93.1%)	27 ^g^	
Wijesekara, N. (2021) [61]	Letter to the Editor	Modelling	Sri Lanka	22 April 2020	Navy sailor attached to Sri Lanka Naval Base at Welisara	Y			
Baettig, S. J. (2020) [26]	Research Article	Case series	Switzerland	12–27 March 2020	Swiss Armed Forces; symptomatic recruit presenting at military medical center of Monte Ceneri and subsequent outbreak monitoring	N			Recruit school in Monte Ceneri
Bielecki, M. (2021) [27]	Research Article	Prospective cohort/Cross-sectional	Switzerland	25 March–3 May 2020	Swiss Armed Forces; symptomatic soldiers presenting at military medical center and asymptomatic soldiers who were sampled cross-sectionally	N	526 (90%)	20.6 (18–54) ^a^	Army base in Airolo
*Crameri, G.A.G. (2020)* [18]	*Rapid Communication*	*Retrospective cohort*	*Switzerland*		*Swiss Armed Forces; recruits of two heavily affected companies with available information on SARS-CoV-2 status*	*N*	*174 (87%)*	*20.7 (19.9 to 21.8)*	*Army base in Airolo*
Handrick, S. (2020) [36]	Short Report	Case series	Tunisia	21 March 2020	Tunisian soldier; arrival testing at Tunis-Carthage Airport	Y	1 (100%)	31	Arrival testing upon return from deployment
Stachow, E. (2021) [56]	Research Letter	Retrospective cohort	United Kingdom		Symptomatic military personnel presenting to Royal Navy medical team	N			
Taylor, H. (2021) [59]	Research Article	Prospective outbreak investigation	United Kingdom	5 weeks from 30 March 2021	All adult Army personnel working in the Army barracks	N			Army barracks
Clifton, G.T. (2021) [29]	CDC MMWR	Cross-sectional	United States	28–30 April 2020	United States Army active duty soldiers deployed to field hospital to provide care to COVID-19 patients	N	201 (59.8%)	32 (25.3–40.0)	Deployment to Javits New York Medical Station
Elliott, B.P. (2021) [32]	Research Article	Retrospective cohort study	United States	1 June–13 November 2020	Active duty servicemen with laboratory-confirmed severe or life-threatening COVID-19 in a military treatment facility, confirmed positive by SARS-CoV-2 reverse transcription polymerase chain reaction (RT-PCR)	Y			Wright-Patterson Medical Center
Kasper, M.R. (2020) [38]	Research Article	Case series	United States	23 March–18 May 2020	USS Theodore Roosevelt sailors under emergency public health outbreak investigation	N	3733 (78.1%)	27.2 (18–59) ^b^	Aircraft carrier
Kim, S.Y. (2021) ^$^ [39]	Research Article	Pandemic response	United States	January–December 2020	United States Forces Korea service members and affiliates	Y			United States Forces Korea Bases
Kline, J.D. (2020) [40]	Research Article	Case report	United States	26 March 2020	United States National Guard mobilized for national guard annual training, presented to emergency department	Y	1 (100%)	36	Winn Army Community Hospital, Fort Stewart, Georgia
Kwon, P.O. (2020) [41]	News Article-Case Series	Case report	United States	9 October 2020	United States Army Officer under quarantine as close contact of case	Y	1 (100%)	34	Quarantine Facility
Lalani, T. (2021) [42]	Research Article	Cross-sectional	United States	8–16 May 2020	United States Navy personnel and other military health care workers deployed on hospital ship	N	217 (50.2%)	18–29 (209/432, 48.4%) 30–39 (120/432, 27.8%) 40+ (103/432, 23.8%)	Deployment on USNS COMFORT
Letizia, A.G. (2020) [44]	Research Article	Cohort	United States	2 weeks from 12 May–15 July 2020	United States Marine Corps recruits under 14-day supervised quarantine before being allowed to enter Parris Island, part of CHARM study	N			Quarantine Facility (hotel/closed college campus) on Marine Corps Recruit Depot, Parris Island
Marcus, J.E. (2020) [45]	CDC MMWR	Cohort	United States	1 March–18 April 2020	United States Air Force trainees under quarantine before basic military training	N			Arrival Quarantine at Air Force Base Joint Base San Antonio-Lackland (JBSA)
Marcus, J.E. (2021) [46]	Research Article	Cohort	United States	11 May–24 August 2020	United States Air Force trainees under quarantine before basic military training	N			Arrival Quarantine at Air Force Base Joint Base San Antonio-Lackland (JBSA)
Servies, T. (2020) [53]	News Article		United States	12 March–17 April 2020	COVID-19 cases in United States active component military personnel in Europe, monitored by U.S. Army Public Health Command Europe (PHCE) on Army Disease Reporting System internet (ADRSi) for Army facilities	Y	66 (83.5%)	20–56 ^d^	
Smith, L. (2020) [55]	Research Article	Case Report	United States	25 March 2020	United States military personnel, active duty male who reported sick	Y	1 (100%)	21	Military sick call clinic
Stanila, V. (2020) [57]	News Article		United States	11 March–30 September 2020	United States active duty service members air evacuated in CENTCOM and EUCOM for COVID-19	Y	204 (90.7%)	<20 (2/225, 0.9%) 20–24 (37/225, 16.4%) 25–29 (55/225, 24.4%) 30–39 (95/225, 42.2%) 40+ (36/225, 16.0%)	Air Evacuation from CENTCOM and EUCOM countries
Ghoddusi, F. (2021) [34]	Research Article	Case report	United States	November 2020	United States Army reservist on active duty orders presenting to outpatient clinic in Kuwait	Y	1 (100%)	28	Outpatient in a role 1 facility (in deployed environment)
Letizia, A.G. (2021) [43]	Research Article	Prospective cohort	United States	11 May–2 November 2020	United States Marine Corps recruits followed up for 6 weeks during training at Marine Corps Recruit Depot-Parris Island (MCRDPI), part of CHARM study	N	2622 (92.0%)	19.1 (1.9)	Marine Corps Recruit Depot Parris Island (MCRDPI)
Sikorski, C.S. (2021) [54]	Research Article	Case report and pandemic response	United States	26 February–27 April 2020	US military population in Italy diagnosed at an Italian military hospital	N			Military hospital in Italy
*Alvarado, G.R. (2020)* [16]	*Research Letter*	*Case series*	*United States*	*31 March* *–15 April 2020*	*USS Theodore Roosevelt sailors who disembarked at Naval Base Guam*	*N*			*Aircraft carrier*
*Kebisek, J. (2020)* [19]	*News Article*	*Special report*	*United States*	*11 February* *–6 April 2020*	*Confirmed cases among United States Army active component service members reported to the DRSi with symptom onset dates from 17 February 2020 through 2 April 2020 from any military treatment facility (MTF)*	*Y*	*175 (79.9%)*	*<25 (34/219, 15.5%)* *25–34 (76/219, 34.7%)* *35–44 (67/219, 30.6%)* *45+ (42/219, 19.2%)*	*Military treatment facilities*
*Payne, D.C. (2020)* [20]	*CDC MMWR*	*Cross-sectional*	*United States*	*20* *–24 April 2020*	*USS Theodore Roosevelt sailors under outbreak investigation, convenience sample of 382 service members serving aboard the aircraft carrier*	*N*	*289 (75.7%)*	*30 (24–35)*	*Aircraft carrier*
*Stidham, R.A. (2020)* [23]	*News Article*		*United States*	*1 January* *–30 September 2020*	*United States active duty service members, recruits, reserve/guard, cadets; confirmed or probable cases in Disease Reporting System Internet (DRSi)*	*Y*	*Active duty (23,987/39,703, 60.4%)* *Recruits (5491/39,703, 13.8%)* *Reserve/Guard (1880/39,703, 4.7%)* *Cadets (326/39,703, 0.8%)*	*Active duty/Recruits/Reserve or Guard/Cadets* *15–19 (2517/39,703, 6.3%)/(4004/39,703, 10.1%)/(332/39,703, 0.8%)/(221/39,703, 0.6%)* *20–24 (11,033/39,703, 27.8%)/(2008/39,703, 5.1%)/(537/39,703, 1.4%)/(205/39,703, 0.5%)* *25–29 (6944/39,703, 17.5%)/(571/39,703, 1.4%)/(432/39,703, 1.1%)/(2/39,703, 0.005%)* *30–34 (3952/39,703, 10.0%)/(176/39,703, 0.4%)/(377/39,703, 0.9%)/0* *35–39 (2777/39,703, 7.0%)/(41/39,703, 0.1%)/(275/39,703, 0.7%)/0* *40–44 (1403/39,703, 3.5%)/(7/39,703, 0.02%)/(193/39,703, 0.5%)/0* *45–49 (760/39,703, 1.9%)/0/(152/39,703, 0.4%)/0* *50–54 (363/39,703, 0.9%)/0/(136/39,703, 0.3%)/0* *55–59 (147/39,703, 0.4%)/0/(55/39,703, 0.1%)/0* *60–64 (54/39,703, 0.1%)/0/(8/39,703, 0.02%)/0* *65+ (20/39,703, 0.05%)/0/(1/39,703, 0.003%)/0*	
*Vick, D.J. (2021)* [25]	*Research Letter*	*Letter*	*United States*	*Through 2 November 2020*	*COVID-19 cases in United States military members reported by Department of Defense*	*Y*			

^α^ Studies italicized are excluded from all analysis due to potential overlap in cases, included in the tables only for presentation purposes, except Talmy et al.: study is included for the meta-analysis of symptoms since it is the only Israel study with symptom-related information. ^&^ Date of case confirmation extracted for case report and outbreak period extracted for others without clear mention of study period. ^@^ Study focuses on active serving military members; only data for this group extracted in papers that also mentioned family, dependents, retirees, as long as breakdown provided, unless indicated otherwise. ^^^ Male (%) and age refer to that of the cases if study only reports on cases (indicated by Y in column “cases only”); otherwise, it is reserved exclusively for data on entire population if study also mentions non-cases.* Case was mentioned in the introduction of the article. ^$^ Cases include non-service members; study on United States Forces Korea includes affiliates, as breakdown was not provided; study on South Korean military includes civilian employees; included because South Korean military regards them as part of military population. ^#^ Sole study with military and police combined in a category, study did not provide breakdown. ^%^ Median (IQR), Mean (SD), unless indicated otherwise; actual age given for case report. ^a^ Median (range). ^b^ Mean (range). ^c^ Range of population mentioned in study: 18–50 years old. ^d^ Range. ^e^ Median (SD). ^f^ Mean of mean (SD) of each unit. ^g^ Median.

#### 3.1.2. COVID-19 Incidence and Demographic of Cases

A total of 8635 cases were reported from a known military population of 711,408 between January 2020 and early May 2021 (Table 2). The majority of the population was attributed to Oh et al., who reported a denominator of 599,000 military personnel constituting both active servicemen and civilian employees in South Korea. In those confirmed to have received COVID-19 testing, positivity by RT-PCR and serology were 5817/51,083 (11.39%) and 81/3538 (2.29%), respectively. Most studies had reported low incidence of cases, and high incidence (>0.50%) was observed in only three studies—Joshi et al., Paleiron et al., and Sasongko et al. We observed a pooled COVID-19 incidence of 0.19% (*n* = 22; 95% CI: 0.00–9.18%), with high heterogeneity among the studies (I^2^: 99.93%) (Figure 2a). There were 3400 (76.73%) males out of 4431 cases with known gender (Appendix A). Race was reported for 1414 cases by five studies from the U.S. (Appendix A). The majority of cases occurred in White Americans (628; 44.41%), followed by Hispanic/Latino Americans (274; 19.38%), African Americans (274; 19.38%), Asian/Pacific Islanders (106; 7.50%), Native Americans (34; 2.40%), and others (98; 6.93%).

Thirteen studies reported rank information for 4332 cases—1418 cases (32.73%) were in the command line (officers and non-commissioned officers), and 2914 cases (67.27%) were in enlisted soldiers (Appendix A). Of cases in enlisted soldiers, 1386 occurred in newly conscripted recruits. Five studies reported information on the job nature of 578 cases—twenty-three (3.98%) cases had medical responsibilities as registered nurses, physician/physician assistants, corpsman, and medical support with patient interaction or medics; five (0.87%) were trainers who returned from a deployment to a military education and training center in Niger. Kasper et al. reported the work departments of 553 cases on board the USS Theodore Roosevelt—supply (139), reactor (138), weapons (94), engineering (67), air (65), combat support division (38), and deck (4); eight cases from the medical division were amongst twenty-three cases mentioned earlier. Incidence of COVID-19 was the highest in weapons (94/226, 41.59%), followed by supply (139/358, 38.82%), engineering (67/204, 32.84%), reactor (138/440, 31.36%), combat support division (38/202, 18.81%), medical (8/48, 16.67%), air (65/459, 14.16%), and deck (4/86, 4.65%). The odds of COVID-19 disease were 1.73 to 2.70 times significantly higher in the weapons, supply, and engineering and reactor divisions with reference to air division, which the study attributed to the divisions’ (reactor, engineering, supply, weapons and combat support) predisposition to more confined working spaces as compared to the air and deck departments.

**Table 2 ijerph-19-07418-t002:** COVID-19 outcomes in study populations.

Study	Outcomes (Not Included in Original Study’s Analysis)			Outcomes (Included in Original Study’s Analysis)							Clinical Characteristics			
	No. of Cases	Total Personnel	Infected (%)	No. of Cases	Total Personnel ^#^	Infected (%)	Testing Coverage ^%^	Testing Mode	RT-PCR	Serology	Symptomatic (%)	Asymptomatic (%)	Hospitalization	Mortality
Pirnay, J.P. (2020) [51]				9	70	12.86%	70	RT-PCR, Serology, and Sequencing	4/9	9/9	5 (55.6%)	4 (44.4%)	0	0
Escalera-Antezana, J.P. (2020) [33]				1261	50,040	2.52%								24
Pasqualotto, A.C. (2021) [50]				52 ^h^	1592	3.27%	1592	RT-PCR, Serology	4/50	52/1592				
Halladay, J. (2020) [35]	1834			55	1700	3.24%								0
Elhakim, M. (2020) [31]				1										
Paleiron, N. (2021) [49]				1279	1688	75.77%	1688	RT-PCR	1038/1688		1107 (86.6%)	172 (13.4%)	107	0
Chassery, L. (2021) [17]				1148							≥50		2	0
Joshi, R.K. (2020) [37]				27	34	79.41%	34	RT-PCR	27/34		0	27 (100%)		
Sasongko, S. (2021) [52]				144	173	83.24%	173	RT-PCR	144/173		144 (100%)	0	144	
Nitecki, M. (2021) [47]	1477	31,005	4.76%	1338	24,362	5.49%	24,362	RT-PCR	1338/24,362		1338 (100%)	0		
*Segal, D. (2020)* [22]				*6*				*rRT-PCR*	*6*					
Talmy, T. (2021) [58]	219			119				RT-PCR	119		119 (100%)	0		0
*Segal, D. (2021)* [21]				*726*	*18,719*	*3.88%*		*RT-PCR*						
*Tsur, A. (2021)* [24]	*237*			*3^j^*	*986*	*0.30%*		*RT-PCR, Serology*			*3 (100%)*	*0*		
Di Nunno, D. (2020) [30]				81							81 (100%)	0	81	
Borud, E.K. (2021) [28]				1	1114 ^b^	0.09%	1114	RT-PCR and Serology	0/1114	1/1114	0	1 (100%)		
Velasco, J.M. (2020) [60]				515	5042	10.21%	5046	rRT-PCR	515/5046					
Oh, H.S. (2020) ^$^ [48]				58	599,000	0.01%					58 (100%)	0		0
Wijesekara, N. (2021) [61]	28			1										
Baettig, S.J. (2020) [26]	3			2	140 ^a^	1.43%	55	RT-PCR and Serology	2/7	2/55	2 (100%)	0	0	0
Bielecki, M. (2021) [27]				255	584	43.66%		RT-PCR and Serology			107 (42.0%)	148 (58.0%)	1	0
*Crameri, G.A.G. (2020)* [18]				*145*	*199*	*72.86%*	*199*	*RT-qPCR, Serology*			*68 (46.9%)*	*77 (53.1%)*		
Handrick, S. (2020) [36]				1				RT-qPCR, Sequencing	1		0	1 (100%)		
Stachow, E. (2021) [56]				21	79	26.58%	79	RT-PCR	21/79		21 (100%)	0		
Taylor, H. (2021) [59]				7 ^i^	254	2.76%	254	RT-PCR, Serology, Sequencing			7 (100%)	0		
Clifton, G.T. (2021) [29]	8	591	1.35%	6	336	1.79%	336	RT-PCR and Serology	2/336	5/336	4 (66.7%)	2 (33.3%)		
Elliott, B.P. (2021) [32]				1 ^c^				RT-PCR	1		1 (100%)	0	1	
Kasper, M.R. (2020) [38]				1331	4779	27.85%	4779	rRT-PCR	1271/4779		759 (57.0%)	572 (43.0%)	23	1
Kim, S.Y. (2021) ^$^ [39]				479										
Kline, J.D. (2020) [40]				1				RT-PCR	1		1 (100%)	0	1	0
Kwon, P.O. (2020) [41]				1				RT-PCR	1		1 (100%)	0	0	0
Lalani, T. (2021) [42]	18	>1200	1.50%	13	432	3.01%	432	RT-PCR and Serology	8/432	12/432	5 (38.5%)	8 (61.5%)		
Letizia, A. G.(2020) [44]	57	3362	1.70%	31^d^	1708^d^	1.81%		RT-PCR and Sequencing						
Marcus, J.E. (2020) [45]				4	4073	0.10%	85	RT-PCR	4/85		4 (100%)	0	0	0
Marcus, J.E. (2021) [46]	273	10617	2.57%	269 ^e^	10,479^e^	2.57%	10479	RT-PCR	269/10479				0	
Servies, T. (2020) [53]	84			79				RT-PCR	79		68 (100%)	0	3	0
Smith, L. (2020) [55]				1				RT-PCR	1		1 (100%)	0	0	0
Stanila, V. (2020) [57]	72,671 as of 10 December 2020 *			225							45 (80.4%)	11 (19.6%)		
Ghoddusi, F. (2021) [34]				1				RT-PCR	1		1 (100%)	0	1	0
Letizia, A.G. (2021) [43]				1079 ^g^	2851 ^g^	37.85%	2319	RT-PCR, Serology	1079/2319		347 (32.2%)	732 (67.8%)	0	0
Sikorski, C.S. (2021) [54]				6				RT-PCR	6					
*Alvarado, G.R. (2020)* [16]				*736*	*4085*	*18.02%*	*4085*	*RT-PCR*	*736/4085*		*590 (80.2%)*	*146 (19.8%)*	*6*	*1*
*Kebisek, J. (2020)* [19]	*328*			*219*	*487,100*	*0.04%*		*RT-PCR*			*219 (100%)*	*0*	*12*	
*Payne, D.C. (2020)* [20]	*235*			*238*	*382*	*62.30%*	*382*	*RT-PCR, Serology*	*98/267*	*228/382*	*194 (81.5%)*	*44 (18.5%)*	*2*	
*Stidham, R.A. (2020)* [23]				*39,703 ^f^*									*586*	*8*
*Vick, D.J. (2021)* [25]				*58,081*										

^#^ Includes all susceptible personnel within population mentioned in study, may not be 100% tested. ^%^ Refers to the number of personnel known to receive testing. * Cases in active duty service members in CENTCOM (492) and EUCOM (2443): Kuwait (171), Afghanistan (79), Iraq (95), Afghanistan (89), Saudi Arabia (41), Qatar (48), UAE (29), Bahrain (29), Germany (1769), Italy (165), Spain (76), Turkey (48), Belgium (44), and Romania (4). ^$^ Cases include non-service members; study on United States Forces Korea includes affiliates as breakdown was not provided; study on South Korean military includes civilian employees; included because South Korean military regards them as part of military population. ^a^ 140 recruits within the concerned company although only 55 were quarantined. ^b^ Cases found positive by rapid test, serology, or PCR on enrollment day excluded. ^c^ Calculated from percentage given in study. ^d^ Minimum number of cases discernible from the study due to loss-to-follow-up from the initial population; denominator of susceptible individuals calculated using all recruits with baseline positive results (1813) – baseline positive (105) = 1708. ^e^ Cases positive on arrival test excluded (*n* = 134). ^f^ Cases include active duty servicemen, recruits, reserves/guards, and cadets. ^g^ Only used population seronegative on enrolment. ^h^ Calculated from percentage given in study, 52 or 53 cases, minimum taken. ^i^ Cases reflected are the minimum discernible; there could be more cases, but breakdown was not provided. ^j^ Cases regarded as true positive using serology test did not report 11 regarded as false positive by PCR.

#### 3.1.3. Deployment and Possible Exposures

Location of possible exposure was reported by 15 studies; a higher incidence was observed for possible exposure overseas (*n* = 3; 39.85%; 95%CI: 0.00–95.87%) as compared to possible local exposure (*n* = 12; 3.03%; 95%CI: 0.00–12.53%) (Figure 2c). Incidence of cases was higher in those who were deployed (*n* = 6; 26.78%; 95%CI: 0.00–71.51%) as compared to those not deployed (*n* = 9; 4.37%; 95%CI: 0.00–17.93%) (Figure 2d). Areas of deployment outside of the cases’ home country include Djibouti, Morocco, South Korea, Niger, Europe, countries in the U.S., Central and Europe Command (CENTCOM/EUCOM), and aircraft carriers (Table 3). The duration of deployment ranged from 1 day to 5 months in seven studies. Pooled incidence of COVID-19 from six available studies was higher among overseas deployment (*n* = 3; 39.84%; 95%CI: 0.00–95.87%) than local deployment (*n* = 3; 3.03%; 95%CI: 2.37–3.74%) (Figure 2e). The three overseas deployments recorded were on aircraft carriers USS Theodore Roosevelt, Charles de Gaulle, and Belgium forces to a military training institute in Niger [38,49,51]. The three local deployments were to Javits New York Medical Station (a field hospital in New York City), USNS Comfort (a U.S. Navy hospital ship), and long-term care facilities in Canada [29,35,42]. Significant heterogeneities (I^2^ = 100%) were observed for all subgroup analyses on the location of exposure and deployment. 

**Table 3 ijerph-19-07418-t003:** Possible COVID-19 exposures reported by cases in included studies (only those with exposures presented here).

Study	Local/Overseas	Deployment	Country of Deployment	Duration of Deployment	Recreational Activity	Close Contact (Secondary Transmission)	Healthcare	Other Exposure
Pirnay, J.P. (2020) [51]	Overseas	Y	Niger	12 December 2019/1 February 2020–13/19 May 2020				Provision of military education and training overseas (9/9, 100%) ^^^
Escalera-Antezana, J.P. (2020) [33]	Local							Capital department, La Paz, highest concentration of forces, highest proportion of infected personnel (53.8%)
Pasqualotto, A C. (2021) [50]	Local	N				Close contact with COVID-19 cases (438/1592, 27.5%) ^@^		
Halladay, J. (2020) [35]	Local	Y		3 months				Worked closely with LTC facility staff to carry out day-to-day operations and support infection control and prevention
Elhakim, M. (2020) [31]	Overseas	Y	Djibouti	1 day				
Paleiron, N. (2021) [49]	Overseas	Y		22 January–13 April 2020				
*Chassery, L. (2021)* [17]	*Overseas*	*Y*	*Brest (13–16 March 2020)*	*21 January*–*8 April 2020; supposedly until 23 April 2020, disease started on 5 April 2020*				*Stopover in Brest for technical needs and change of crew*
Joshi, R.K. (2020) [37]		N			Return from leave (27/27, 100%) ^&^	Secondary transmission from asymptomatic cases in cohort (≥7/27)		
*Segal, D. (2020)* [22]	*Local*	*N*						
Di Nunno, D. (2020) [30]	Overseas	Y						
Borud, E.K. (2021) [28]	Local	N						
Oh, H.S. (2020) [48]						Close contact with military member (28/58, 48.3%); Close contact with local community (27/58, 46.6%)	Healthcare associated (2/58, 3.4%)	Unknown (1/58, 1.7%)
Wijesekara, N. (2021) [61]					On leave (1/1, 100%)			
Baettig, S.J. (2020) [26]	Local	N			Vacation trip (1/2, 50%)	Second case might be secondary transmission (1/2, 50%)		
Bielecki, M. (2021) [27]	Local	N						
*Crameri, G.A.G. (2020)* [18]	*Local*	*N*						
Handrick, S. (2020) [36]	Overseas	Y	Morocco					
Clifton, G.T. (2021) [29]	Local	Y		24 March–30 April 2020	Travel within 2 weeks before arrival (0/6, 0%)		Direct care for COVID-19 patients (2/6, 33.3%) Break in PPE (1/6, 16.7%) Aerosol generating procedure (0/6, 0%)	Median direct patient care hours: 264 (228–300)
Kasper, M.R. (2020) [38]	Overseas	Y		13 days				1. Crew working in tighter spaces (e.g., reactor (1.73 (1.29–2.36)), engineering (1.85 (1.29–2.67)), supply (2.41 (1.78–3.26)), and weapons (2.70 (1.92–3.8)) departments) appeared more likely to have confirmed or suspected COVID-19 than those working in a combination of open-air and confined conditions (e.g., air and deck crew). 2. Members of the medical department, who wore personal protective equipment when evaluating crew members, had a somewhat lower attack rate (16.7% 8 cases among 48 personnel) than the overall crew despite being at highest risk as a result of exposure to patients with COVID-19 in a small space.
Kim, S.Y. (2021) [39]	Overseas	Y	South Korea					
Kline, J.D. (2020) [40]	Local	N						Resident of an area that reported a cluster of COVID-19 cases, mobilization for national guard training, and recently traveled to Fort Stewart, Bartow County, Georgia
Kwon, P.O. (2020) [41]	Overseas	Y				Close contact of confirmed case (1/1, 100%)		
Lalani, T. (2021) [42]	Local	Y		28 March–30 April 2020		Direct interaction with COVID-19 patients/individuals 2 weeks before deployment: No (1/13, 7.7%); Do not know (3/13, 23.1%); Yes (9/13, 69.2%). Anyone in workspace/berthing/social circle placed in isolation/quarantine: No (1/13, 7.7%); Do not know (3/13, 23.1%); Yes (9/13, 69.2%).	Primary workspace during deployment being ICU/Ward (12/13, 92.3%) Direct care of COVID-19 patients during deployment (12/13, 92.3%) Spent 2/3 or more of time in direct patient care during deployment (11/13, 84.6%) Aerosol generating procedure (5/13, 38.4%)	Berthing during deployment: Enlisted berthing (0/13, 0%); Non-government organization berthing (1/13, 7.7%); Officer berthing (1/13, 7.7%); Private hotel room (11/13, 84.6%). Place where meals were consumed: Galley (2/13, 15.4%); Hotel room (8/13, 61.5%); Other (1/13, 7.7%); Workspace (2/13, 15.4%).
Letizia, A.G. (2020) [44]	Local	N						Had an infected roommate (24/77, 31.2%) ^$^
Marcus, J.E. (2020) [45]	Local	N				Contact of patient A during training (3/4, 75%)		First case speculated to have been infected during transit because he arrived from a state not reporting community spread of COVID-19 (1/4, 25%)
Marcus, J.E. (2021) [46]	Local	N						
Servies, T. (2020) [53]	Overseas	Y	Europe					
Smith, L. (2020) [55]	Local	N			Recent recreational activities (clubs, beach) (1/1, 100%)			Work in busy environment (1/1, 100%)
Stanila, V. (2020) [57]	Overseas	Y	Kuwait, Saudi Arabia, Afghanistan, Iraq, Poland, Jordan, Kosovo ^%^					
Ghoddusi, F. (2021) [34]	Overseas	Y	Kuwait					
Letizia, A.G. (2021) [43]	Local	N						
Sikorski, C.S. (2021) [54]	Overseas	Y	Italy					
*Alvarado, G.R. (2020)* [16]	*Overseas*	*Y*						
*Payne, D.C. (2020)* [20]	*Overseas*	Y	*Western Pacific*	*Mid-January*–*End March*		*Reported contact with known COVID-19 case (64.2%) compared with those who did not (41.7%) (OR = 2.5; 95% CI = 1.1–5.8)* *Sharing the same sleeping berth with a crewmember who had positive test results (65.6%) compared with those who did not (36.4%) (OR = 3.3; 95% CI = 1.8–6.1)*		*Reported prevention behaviors:* *Increased hand washing (218/238, 91.6%) OR: 0.90 (0.42–1.94);* *Hand sanitizer use (219/238, 92.0%) OR: 0.59 (0.24–1.44);* *Avoiding common areas (78/238, 32.8%) OR: 0.56 (0.37–0.86) ^§^;* *Face covering use (158/238, 66.4%) OR: 0.30 (0.17–0.52) ^§^;* *Increased workspace cleaning (195/238, 81.9%) OR: 1.30 (0.78–2.16);* *Increased berthing cleaning (156/238, 65.5%) OR: 0.95 (0.61–1.47);* *Increased distance from others (105/238, 44.1%) OR: 0.52 (0.34–0.79)^§^*

^$^ Includes cases who were not participants of the study. ^&^ All cases were under quarantine after return from leave but at least 7 cases confirmed to be from secondary transmission within the facility. ^@^ Median time for exposure to COVID-19 cases was 21 days before study participation. ^%^ U.S. Central Command (CENTCOM) and U.S. European Command (EUCOM) countries. ^§^ Significant *p*-Value reported by original study analysis. ^^^ Belgian military servicemen were infected by a locally transmitted virus with a recent African common ancestor, as discovered from sequencing result.

A plausible transmission origin was recorded for 579 cases in 12 studies (Table 3). Thirty cases (5.18%) had recent recreational activity before testing positive, such as a vacation trip, visited clubs/beaches, or returned from leave, while 538 cases (92.92%) were reportedly contacts. Twenty-four cases, specifically, had infected roommates. Belgium military personnel were found to be infected with a strain with a recent African common ancestor, indicating local transmission during their deployment to Niger [51]. Separately, there were cases that were likely infected by the public during transit or in their residential area [40,45]. Possible healthcare-associated exposure was reported in at least 16 cases (2.76%) by three studies—14 cases were directly involved in the care of COVID-19 patients, five of whom had conducted aerosol generating procedure, and one had reported a break in personal protection equipment [29,42,48].

#### 3.1.4. Clinical Characteristics

Of twenty-five studies reporting the clinical characteristics of cases (Appendix A), we observed a pooled incidence of 77.90% (*n* = 16, 95%CI: 43.91–100.00%) symptomatic cases with significant heterogeneity among the studies (I^2^: 99.51%) (Figure 3a). Several studies reported a high incidence of asymptomatic subjects ranging from 58.04% to 100.00% [27,37,42,43]. Most cases developed mild disease; pooled incidence of hospitalization was 4.43% (*n* = 9; 95%CI: 0.00–25.35%) among cases (Figure 3b) and pooled mortality rate 0.25% (*n* = 10; 95%CI: 0.00–0.85%) (Figure 3c). Heterogeneity among the studies was high for both pooled hospitalization rate and mortality rate at 99.57% and 83.64%, respectively. Symptoms with the highest incidence were headache (*n* = 7; 60.78%; 95%CI: 44.43–76.53%), anosmia (*n* = 4; 53.43%; 95%CI: 25.33–80.94%), ageusia (*n* = 3; 43.07%; 95%CI: 16.21–71.11%), anosmia/ageusia (*n* = 3; 43.05%; 95%CI: 18.27–68.81%), myalgia (*n* = 6; 42.95%; 95%CI: 10.20–77.69%), nasal congestion (*n* = 4; 42.42%; 95%CI: 14.66–71.59%), and cough (*n* = 6; 41.10%; 95%CI: 12.11–71.98%). The remaining six symptoms had a pooled incidence lower than 40%—fatigue, dyspnea, rhinorrhea, sore throat, fever, and diarrhea. All forest plots representing incidence of symptoms can be found in Figure 4a–m.

## 4. Discussion

Potential risk factors identified from this analysis include confined working spaces aboard ships, performing aerosol-generating procedures during COVID-19 missions, and having vacations. Payne et al. further identified greater odds of COVID-19 in those who had (1) contact with known cases (OR: 2.5; 95%CI: 1.1–5.8) and (2) shared sleeping berth with cases (OR: 3.3; 95%CI: 1.8–6.1) as compared to those without [20]. These factors coalesced into close physical proximity favoring the spread of respiratory diseases commonly transmissible via airborne or droplet routes, which aligned with evidence showing that communication lasting at least 30 min and sharing a bedroom were associated with higher risk of transmission among household contacts [62]. Medical procedures such as tracheotomy, non-invasive ventilation, and manual ventilation prior to intubation emit respiratory particles [63]. A systematic review found 6.60 times greater odds of SARS-CoV-1 in healthcare personnel who had exposure to aerosol-generating procedures and an absolute increase of 10–15% in risk of SARS-CoV-1 transmission [64]. As for COVID-19, 10.7% of healthcare workers reported suspected or confirmed diagnosis within 32 days of conducting tracheal intubation [65].

Despite heightened susceptibility to disease transmission, we observed low incidence of COVID-19 infection, hospitalization, and mortality of cases in the military population. This was not surprising given that their healthy base profile might have been protective against infection and clinical progression. The military population was young and predominately male (most studies reported a mean age between 20–30 years old), and intakes were heavily screened in terms of medical fitness [47]. Furthermore, the vulnerability of militaries to infectious diseases was compensated by swift, strict, and regimented containment strategies to which the compliant community adhered [22,39,48,54]. As of 3 March 2020, the PLA had reported no confirmed cases due to successful and strengthened prevention measures [66]. The USFK, United States Forces in Italy, South Korean military, and Israel Defense Forces echoed early curtailing of soldiers’ lifestyles and movement, development of extensive testing and contact tracing capacities, and stringent quarantine standards since the early stages of the pandemic [22,39,48,54]. Algorithms guided systematic and efficient flow of response, from screening to immediate quarantine of units with confirmed cases and identification of hotspots on post and close contacts. Additionally, greater access to healthcare services could have contributed to low mortality since military personnel typically receive healthcare services within the military installation/framework and are entitled to comprehensive medical coverage in most countries [67].

A higher pooled incidence of COVID-19 was observed for populations (1) with possible exposure overseas, (2) on deployment, and particularly, (3) overseas deployments in the early phase of the pandemic. This corresponded with higher incidence of disease infection during deployment historically. Higher rates of food borne disease outbreaks were recorded in French soldiers deployed overseas (26.7 outbreaks/100,000) as compared to those in France (2.4 outbreaks/100,000) from 1999 to 2009 [11]. Separately, respiratory illness was newly reported in 14% of deployed soldiers to Iraq and Afghanistan as compared to 10% of non-deployed soldiers [68]. Heightened infection risk during deployment could be due to harsh conditions in foreign environments, possible inadequacy in hygiene and sanitation, sleep deprivation, and physical and mental strain [11]. Amid uncertainties surrounding the novel virus during early stages of the outbreak, these stressors could have synergistically altered the physiological and immunological states of the military, increasing their vulnerability to infections [11]. Airborne and droplet transmission of viruses are likely to be exacerbated by congregated living arrangements and impaired hygiene practices [69]. Interaction with the local populace and environment [69] can further obscure infection control in foreign land due to dependence on local policies and community response, exposing soldiers to the risk of acquiring disease circulating in the local community [51]. The disparity in COVID-19 incidence between local and overseas deployment in this study could be due to differences in deployment purpose and time period. Studies on overseas deployment—USS Theodore Roosevelt, Charles de Gaulle, and Niger military institute—had commenced prior to or in the early stage of the pandemic when COVID-19 information was sparse and received little attention. It was likely that prevention may have been neglected in view of a higher perceived risk from rarer diseases [70], such as diseases endemic to the deployed setting. Chassery et al. also noted that Charles de Gaulle had a stopover in Brest for crew changeover and technical purposes [17]. On the other hand, studies on local deployments were to field hospital, hospital ships, and long-term care facilities during the first wave of the pandemic. The preparedness behind deployments to support medical and care facilities was likely more robust in terms of logistic supply of personal protective equipment, protocol, and heightened awareness of precautionary hygiene measures. 

Contrarily, service members may also act as vessels and introduce viruses to the deployed region. The transmission of pathogens across geographical boundaries was well-documented by Zemke et al. in 67 studies dating between 1955 and 2018 [71]. Just as the first influenza A(H1N1)pdm09 cases in Kuwait and Iraq were imported by U.S. military personnel who were infected during pre-deployment trainings [71,72], Nepalese peacekeepers infected from their mission in Kathmandu served as the source of a major cholera outbreak in Artibonite River Valley in 2010 and the subsequent endemicity of the pathogen in Haiti [71,73]. Likewise, COVID-19 cases have been detected in U.S. military personnel after arriving in South Korea and Japan [74].

Understanding the common clinical traits of COVID-19 in military personnel can improve detection of infected personnel. While the high incidence of symptomatic patients observed in this study aligned with other studies [75,76], the significant proportion of asymptomatic cases could have resulted from scheduled testing practices and social distancing [27,44]. Fever (78%), cough (57%), fatigue (31%), and hyposmia (25%) were the most prevalent symptoms in patients with a mean age of 49 years old from nine countries [77], while fever (46%), cough (37%), and diarrhea (19%) were the most commonly reported in pediatric patients [78]. In this study, headache was the most prevalent symptom, followed by anosmia, ageusia, myalgia, nasal congestion, and cough. It is possible that active testing in the military enabled identification of non-distinctive symptoms such as headache, which are more likely to be overlooked as compared to fever and other respiratory symptoms. The association between headache and anosmia/ageusia and their likely occurrence in the early stage of the disease has been well-established beyond this study [79,80]. Alternatively, differences in disease presentation could plausibly be attributed to infection with different strains that surfaced over the pandemic course. In the United Kingdom, the proportion of asymptomatic cases did not change significantly with the increasing incidence of B.1.1.7 variant [81]. Elsewhere, associations between B.1.1.7 and hospitalization and death were mixed [82]. The delta variant, on the other hand, showed 2.61 times and more than 2 times higher risk of hospitalization compared to alpha variant in England and Scotland, respectively [83].

### 4.1. Preventive Measures

With resonance in the scientific community on the possible endemicity of SARS-CoV-2, vaccination and regular testing as cardinal components of military strategies can facilitate the near future management of COVID-19. The effectiveness of vaccination in military was evident in Singapore during the 2009 H1N1 vaccination program, where the incidence rate was reduced by 54% compared to the unvaccinated prediction [84]. While Pfizer-BioNTech’s vaccine was approved in December 2020, the impact of vaccination against COVID-19 in the military was beyond the focus of this study. Nevertheless, there is sound evidence backing the efficacy of vaccination, inducing protection against infection and reduction in the severity of COVID-19 [85,86]. Vaccination can alleviate the impact of COVID-19 infections resultant of high exposure, preserving military strength and operational readiness. This ought to be complemented with regular testing to identify asymptomatic cases and individuals in their early phase of illness, allowing early intervention to break the transmission chains. In April 2021, the U.S. military reported that one percent of service members in priority groups will be tested fortnightly as testing capacity ramps up [87]. The Singapore Armed Forces has also made adaptations to operate in a COVID-19 endemic environment with regular testing of frontline and service personnel and entry tests for all military events [88,89]. Rapid antigen tests, oropharyngeal/middle turbinate, and deep-throat saliva testing were trialed for adoption in the surveillance program [88,89].

### 4.2. Limitations

While the meta-analyses served as a statistical indicator of our findings, the results were inconclusive due to significant heterogeneity of studies and limited reporting of data by the militaries. The high heterogeneity across all meta-analyses was likely because studies encapsulated different time points of the pandemic geographically and involved different fractions of the military population in varied settings. As Oh et al. reported COVID-19 cases including civilian employees as the full military strength of South Korea [48], we could not discern if this inclusion was consistently practiced worldwide. Similarly, Escalera-Antezana et al. reported infections in terms of all military personnel registered in Bolivia at an unspecified time point [33]. There was a lack of uniformity in included studies, as we attempted to capture the maximum amount of literature related to COVID-19 infections in the military to provide a baseline incidence. Most studies (*n* = 27) had figures cut-off before June 2020, some by December 2020 (*n* = 13), and only a single study was from 2021. The incidence may not be generalizable to the current situation, which has worsened with recurrent waves caused by variants of concern. We also note the saturation of literature published by the developed nations, predominately the United States. The varied degree of transparency exercised by the military worldwide may have caused reporting bias and skewed representation of COVID-19 incidence. The true impact of COVID-19 on military population was further obscured by publication bias, as authors could not discern if the lack of relevant publication identified from the Chinese database (*CKNI*) and English databases was due to true absence of cases attributed to effective prevention control or publication bias. 

Another limitation is the soldier’s autonomy and different testing practices. The accuracy of incidence concluded by individual studies in the early phase of the pandemic could have been diminished as soldiers could refuse sample taking or data usage [28]. While testing rigor was low and typically reserved for symptomatic individuals with defined exposure [29], testing frequency varied across setting depending on risk sentiment, needs, and accessibility to test kits [90,91]. In the U.S. Air Force basic military trainees, the testing prerequisite changed from symptomatic individuals with possible exposure to being symptomatic with mere presentation of symptoms [45] and eventually to universal testing in subsequent cohorts enrolled between May and August 2020 [46]. Given the likelihood of exposure from deployments, it is conceivable that incidence of COVID-19 was heavily underestimated in the early phase of the pandemic.

Lastly, studies were not assessed for quality and publication bias. Since the study types were not restricted, the authors did not deem fit to conduct a quality assessment on literatures that were not intended to be nor held to the rigor of a full-fledged investigational study. The paucity and high heterogeneity of studies investigating COVID-19 outbreaks within the military highlights the need for more attention on this essential but highly susceptible population.

## 5. Conclusions

Despite the low pooled incidence of hospitalization and mortality rates, which were likely attributed to the young, healthy demographic of the military, there was high pooled incidence of symptomatic cases. Active SARS-CoV-2 surveillance strategies is critical for early detection and containment to reduce risk of transmission during military deployments at the early phase of the pandemic. 

## Figures and Tables

**Figure 1 ijerph-19-07418-f001:**
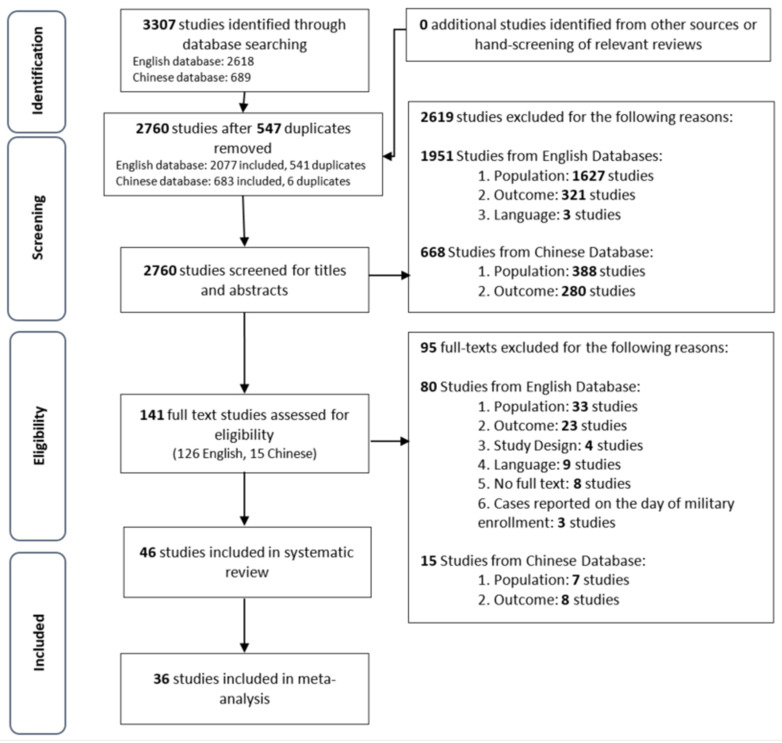
PRISMA flowchart of study selection.

**Figure 2 ijerph-19-07418-f002:**
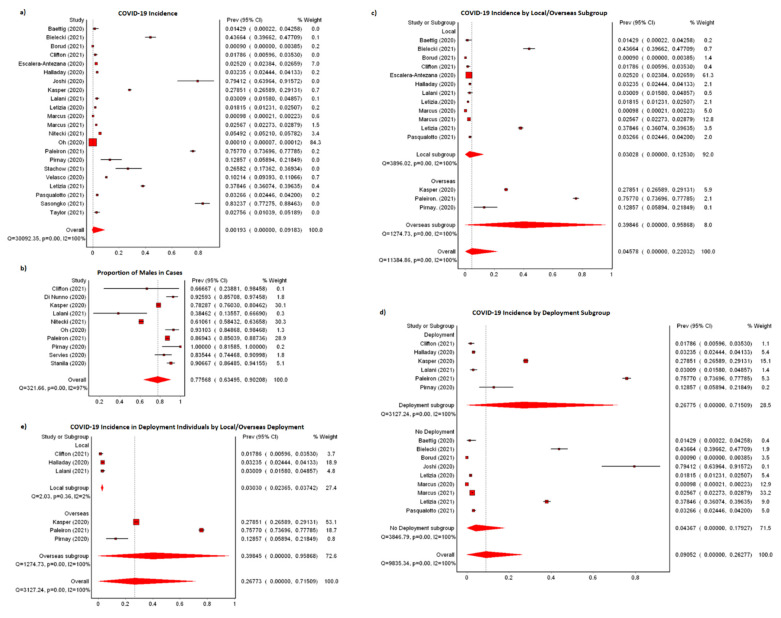
Meta-analyses of COVID-19 incidence (**a**) COVID-19 in military populations, (**b**) COVID-19 incidence by populations with possible exposure locally or overseas, (**c**) COVID-19 incidence by deployment, (**d**) COVID-19 incidence by local or overseas deployment, (**e**) males in cases (using maximum case numbers provided by studies that were not included in original study analysis).

**Figure 3 ijerph-19-07418-f003:**
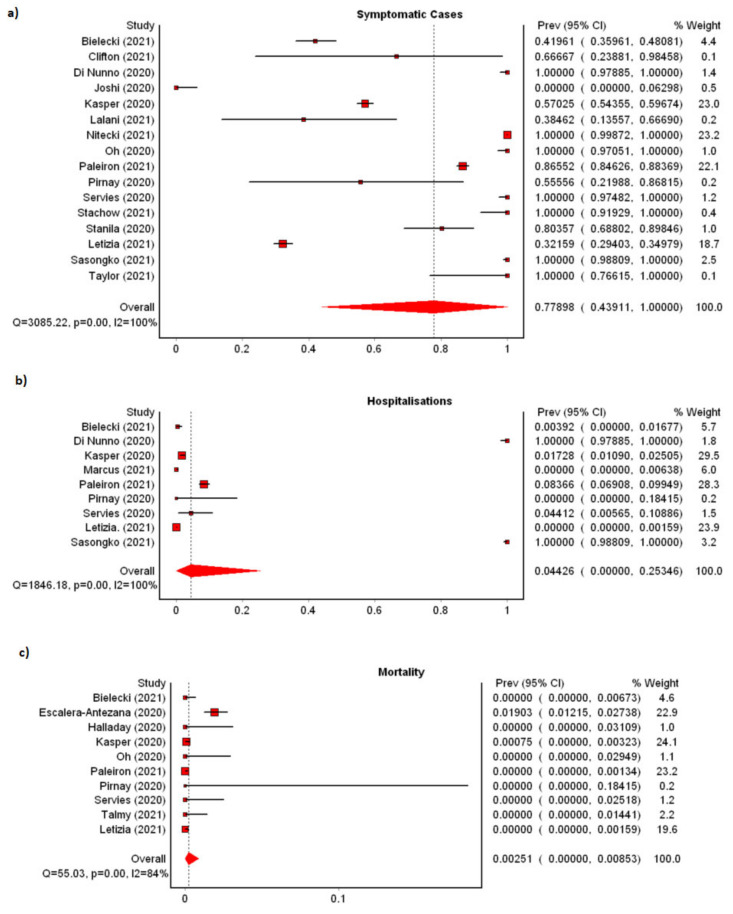
Meta-analyses of incidence of clinical characteristics in cases: (**a**) symptomatic cases, (**b**) hospitalization in cases, (**c**) deaths in cases.

**Figure 4 ijerph-19-07418-f004:**
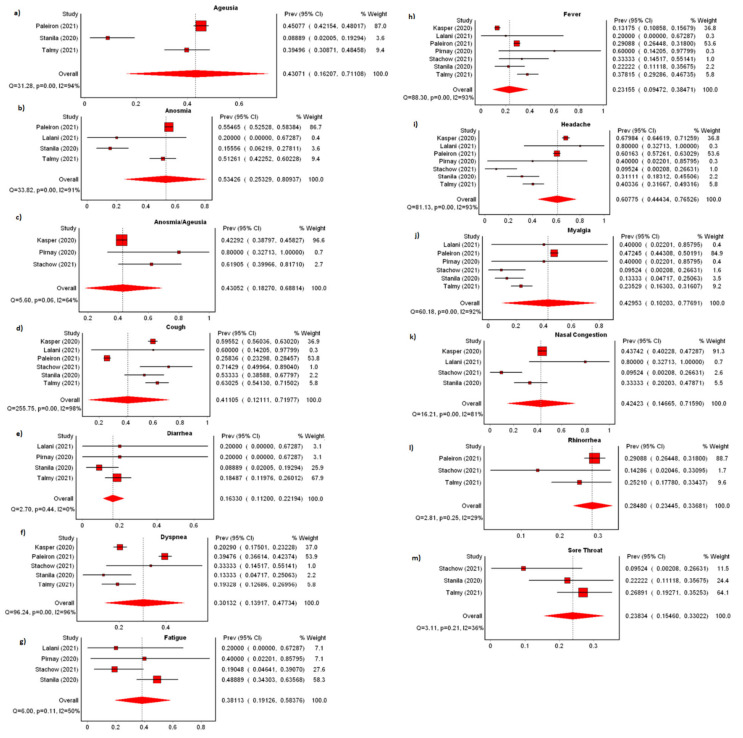
Meta-analyses of incidence (**a**) ageusia, (**b**) anosmia, (**c**) ageusia/anosmia, (**d**) cough, (**e**) diarrhea, (**f**) dyspnea, (**g**) fatigue, (**h**) fever, (**i**) headache, (**j**) myalgia, (**k**) nasal congestion, (**l**) rhinorrhea, (**m**) sore throat.

## Data Availability

The data presented in this study are available in Appendix A.

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
