# Peer review of "SARS-CoV-2 Transmission in the Military during the Early Phase of the Pandemic—A Systematic Analysis"

_ijerph, 2022, doi:10.3390/ijerph19127418_

Round 1

Reviewer 1 Report

Comments to the authors:

Summary: The paper aims to estimate the incidence of SARS-CoV-2 infections in militaries during the early phase of the pandemic, identify potential risk factors behind transmission and their key clinical characteristics. The findings state that there was high pooled incidence of symptomatic cases despite the low pooled incidence of hospitalization and mortality rates, which were likely attributed to the young, healthy demographic of the military. I recommend the publication of this article after consideration of a few minor comments below.

  1. Can the authors please check the whole manuscript for spelling and punctuation errors.

Can the authors elaborate on the rationale and significance in a bit more detail.

Author Response

Dear reviewer, thank you for your kind comments. We have done a proof read of the manuscript for spelling and punctuation as advised. Additionally, we have amended lines 60-67 to emphasize on the significance of this analysis. “While the occurrence of disease outbreaks in military due to overseas deployment and disaster management is well-established [11], the impact and epidemiological characteristics of transmission during the early phase of the SARS-CoV-2 pandemic is not well-studied. Assessing the military’s risk during this pandemic could be a useful reference for novel respiratory virus outbreaks in the future and play a role in guiding their preparation. This study aims to estimate the incidence of SARS-CoV-2 infections in militaries during the early phase of the pandemic, identify potential risk factors behind transmission and their key clinical characteristics.”.

Reviewer 2 Report

It has been more than two years since the first case of COVID-19 in China. The world witnessed five waves of infections, or six at moment, and more upcoming is expected, whereas several challenges are presented. In this periods we witness the virus effects and it’s clear that they do vary between all of this periods. The severity of the disease, differences in symptoms, attitudes of the people have been reported. Differences in the people sense of danger toward the pandemic gradually decreases in most countries mainly due to the approved vaccines, heath policies to control and deal with the disease, and because of the observed dropping in mortality rates.

In this manuscript, the authors reviewed the incidence of COVID-19 in the military, identified the clinical features of COVID-19 cases, and identified possible risk factors for the disease so the manuscript only includ articles through 28 May 2021 (at the beginning of the 3thr wave) and I think that the two last waves are a completely new pandemic reality and it would be interesting to  analyse all of the data including these periods.

In my view, it would be more useful to compare the incidence of COVID_19 in the military with the general incidence (in civilians) in the country of origin (we have access to data from all countries in different time periods) or data from countries were militaries were positioned. 

 You identify these limitations “The high heterogeneity across all meta-analyses was likely because studies encapsulated different time points of the pandemic geographically and involved different fractions of the military population in varied settings”  The study design should include greater homogeneity in the variable "period of time" considering the two COVID-19 waves to discuss the results.

Despite all the limitation that were identified by you in your work, it is always important to discuss pandemics epidemiological aspects in this particular group of the population.

Author Response

Dear reviewer, thank you for your insightful suggestion. Indeed, the most recent waves of COVID-19 were different from the initial waves in terms of differing pandemic response by countries, changes in risk perception and behaviour of the population, and pharmaceutical interventions available. Given evolution of the pandemic course, we have decided to scope this analysis of published literature to the early phase of the pandemic where reliance on non-pharmaceutical interventions was more realistic and much less on vaccination to investigate the early risk faced by military amidst uncertainty of a novel respiratory virus outbreak when a safe and effective vaccine was not ready yet. We believe that assessing the risk faced by militaries specifically in the early phase of outbreak is important as militaries were often the first to be mobilised for crisis management, far before deeper understanding and effective countermeasures towards the pathogen are developed. Inclusion of the later waves should constitute a separate analysis as countries adopted the endemicity route amidst the circulation of newer, more contagious variants, and waned off prevention countermeasures. The vast difference in context and factors should be discussed separately to improve the focus of the paper.

Furthermore, we concur that it will be ideal to compare the incidence of COVID-19 in the military with incidence in the general population. However, information published by militaries worldwide was incredibly fragmented; cases recorded over a few days for a certain country to up to months in another; cases from a specific unit to the entire military; some countries were reported by multiple studies covering different units/time period; some did not have a denominator of the military population to enable calculation of incidence. We will not be able to yield meaningful analysis for any single country in the absence of consistent and regular data over a prolonged time period as we cannot identify specific time frames to make comparison against. The time frames compared for each countries will also differ. We sincerely acknowledge that comparison with incidence in the general population could provide useful insights but perhaps should rely on a different study design in a separate analysis altogether, i.e., usage of governmental sites with periodic updates of COVID-19 cases in the military, if available, and with known denominator of the military strength, instead of published literature. In such analysis, the focus can also be narrowed to some selected countries for exploration in greater depth.

We sincerely hope for your consideration of our explanation.

Reviewer 3 Report

The work presented analyses different variables involved in the transmission of SARS-CoV-2 in the military population during the first months of the pandemic. The authors have carried out a systematic review of the literature, accompanied by an approach to meta-analysis of some sources. As the authors indicate, there were some limitations for the meta-analysis due to the heterogeneity of the studies when analysing the different data collection methodologies carried out by other authors.

As minor observations:

- Page 2, line 75: the keywords searched for in the CKNI database are reflected. Are they the same keywords that were used in the other databases?

- Page 3, figure 1: in some cases, when specifying the reason for exclusion, english and chinese studies are mentioned. It is not clear if this classification refers to the language or the country of origin of the source.

- Page 9, line 417: the format of the references must be revised to adapt to the recommendations of the journal.

- Some bibliographic citations are highlighted. It may have been an error when uploading the file to the platform, but it should be reviewed.

Best regards

Author Response

As minor observations:

- Page 2, line 75: the keywords searched for in the CKNI database are reflected. Are they the same keywords that were used in the other databases?

Dear reviewer, the keywords used for the Chinese database are synonyms of COVID-19, SARS-CoV-2, soldier, and army to fit the search objective of this review. Synonyms were used to identify as many articles using different terms of the same meaning as much as possible.

- Page 3, figure 1: in some cases, when specifying the reason for exclusion, english and chinese studies are mentioned. It is not clear if this classification refers to the language or the country of origin of the source.

Dear reviewer, we would like to clarify that the studies were not excluded because they were English or Chinese. The header was simply to present the screening outcomes of studies from the English and Chinese databases separately. Apologies for the confusion, we have revised the PRISMA flowchart with the heading “studies from XX databases” to reflect this more clearly. Studies excluded under language were those apart from English and Chinese. Hope this clarifies.

- Page 9, line 417: the format of the references must be revised to adapt to the recommendations of the journal.

Dear reviewer, thank you for pointing this out. We have revised the references to MDPI format.

- Some bibliographic citations are highlighted. It may have been an error when uploading the file to the platform, but it should be reviewed.

Dear reviewer, we have reviewed but we are not too sure which citations you are referring to. We have referenced some news articles in the introduction section to provide some realistic examples.

Reviewer 4 Report

In this study, authors have attempted to probe the prevalence of military infection of SARS-CoV-2 at the initial stage of the pandemic. Through the analytic studies, the authors defined limited working spaces aboard ships, aerosol-producing performance, and vacations as representative potential risk factors. The authors have addressed a conclusive note that thorough and continued strategies for SARS-CoV-2 surveillance are critical for prompt detection and subsequent reduction of danger factors during military deployments in the initial phase of pandemic. This manuscript is in general well written and may contain a worth addressing a significance of the preventive public-health measures against SARS-CoV-2 epidemics. I have a few additional comments.

- Use of the word “review” in the text is manifold, although this article seems to be “original article”. The title also contain the word (A Systemic Review). What is the characteristics of the manuscript, review or original article? If authors consider current article to be an original or research article, the word “review” in the entire text is suggested to be minimized. For example, “A Systemic Analysis” instead of “A Systemic Review” in line 3 can be considered.

- The authors may wish to include additional Figure showing representative quantitatively analyzed data of, for example graphs for incidence rates among different cases, in the main body of the manuscript instead of supplementary materials.

Author Response

Comments and Suggestions for Authors

In this study, authors have attempted to probe the prevalence of military infection of SARS-CoV-2 at the initial stage of the pandemic. Through the analytic studies, the authors defined limited working spaces aboard ships, aerosol-producing performance, and vacations as representative potential risk factors. The authors have addressed a conclusive note that thorough and continued strategies for SARS-CoV-2 surveillance are critical for prompt detection and subsequent reduction of danger factors during military deployments in the initial phase of pandemic. This manuscript is in general well written and may contain a worth addressing a significance of the preventive public-health measures against SARS-CoV-2 epidemics. I have a few additional comments.

Dear reviewer, thank you for your kind words.

- Use of the word “review” in the text is manifold, although this article seems to be “original article”. The title also contain the word (A Systemic Review). What is the characteristics of the manuscript, review or original article? If authors consider current article to be an original or research article, the word “review” in the entire text is suggested to be minimized. For example, “A Systemic Analysis” instead of “A Systemic Review” in line 3 can be considered.

Dear reviewer, thank you for the kind suggestion. We agree that this study should be an original article comprising original analysis, as such, we have removed all usage of the term “review” in the manuscript.

- The authors may wish to include additional Figure showing representative quantitatively analyzed data of, for example graphs for incidence rates among different cases, in the main body of the manuscript instead of supplementary materials.

Dear reviewer, thank you for the kind suggestion. We have shifted the 2 figures containing all the forest plots into the main text from the supplementary. They are inserted following their first mention under the relevant subheadings. Hope this suffices.

Round 2

Reviewer 2 Report

Dear authors,

I am satisfied with the your answer  and I consider that the manuscript obtained considerable improvements.